# The Inflammatory Signals Associated with Psychosis: Impact of Comorbid Drug Abuse

**DOI:** 10.3390/biomedicines11020454

**Published:** 2023-02-04

**Authors:** Jesús Herrera-Imbroda, María Flores-López, Paloma Ruiz-Sastre, Carlos Gómez-Sánchez-Lafuente, Antonio Bordallo-Aragón, Fernando Rodríguez de Fonseca, Fermín Mayoral-Cleríes

**Affiliations:** 1Unidad de Gestión Clínica de Salud Mental, Instituto de Investigación Biomédica de Málaga (IBIMA), Hospital Regional Universitario de Málaga, 29010 Málaga, Spain; 2Facultad de Medicina, Universidad de Málaga, Andalucía Tech, Campus de Teatinos s/n, 29071 Málaga, Spain; 3Departamento de Farmacología y Pediatría, Universidad de Málaga, Andalucía Tech, Campus de Teatinos s/n, 29071 Málaga, Spain; 4Facultad de Psicología, Universidad de Málaga, Andalucía Tech, Campus de Teatinos s/n, 29071 Málaga, Spain

**Keywords:** psychosis, addiction, neuroinflammation, NFκB, PPARγ

## Abstract

Psychosis and substance use disorders are two diagnostic categories whose association has been studied for decades. In addition, both psychosis spectrum disorders and drug abuse have recently been linked to multiple pro-inflammatory changes in the central nervous system. We have carried out a narrative review of the literature through a holistic approach. We used PubMed as our search engine. We included in the review all relevant studies looking at pro-inflammatory changes in psychotic disorders and substance use disorders. We found that there are multiple studies that relate various pro-inflammatory lipids and proteins with psychosis and substance use disorders, with an overlap between the two. The main findings involve inflammatory mediators such as cytokines, chemokines, endocannabinoids, eicosanoids, lysophospholipds and/or bacterial products. Many of these findings are present in different phases of psychosis and in substance use disorders such as cannabis, cocaine, methamphetamines, alcohol and nicotine. Psychosis and substance use disorders may have a common origin in an abnormal neurodevelopment caused, among other factors, by a neuroinflammatory process. A possible convergent pathway is that which interrelates the transcriptional factors NFκB and PPARγ. This may have future clinical implications.

## 1. Introduction

Psychosis is generally defined as a mental phenomenon of a pathological nature in which the affected subject experiences a loss of contact with reality. In classical psychiatry, “psychosis” was conceptualized as a large syndrome opposite to “neurosis”, in the form of a characteristic structure that begins to form in childhood and hatches and develops at the beginning of adulthood. Today, psychosis is considered as a common and functional set of symptoms that are present in many psychiatric, neurological, medical and neurodevelopmental conditions [1].

Despite the advances made in the field of the study of the causes and factors involved in the pathogenesis of psychosis, there are still unresolved questions; thus, it remains an old aspiration of biological psychiatry to find biomarkers that can serve as diagnostic criteria for the different disorders of the psychotic spectrum. A new approach to the phenomenon is to “deconstruct” psychosis in its clinical dimensions to identify the pathogenic causes and mechanisms of each of its symptoms and psychopathological dimensions. Thus, we will be able to understand their interaction in individual cases [2].

Why does psychosis occur? This is one of the questions that researchers continue to ask themselves around the world. Although more and more light has been shed on the biological processes that lead a subject to lose touch with their surrounding reality, the truth is that there is still a long way to go. The classical hypothesis that explains the development of psychotic disorders, taking schizophrenia as a paradigmatic example, is the dopaminergic hypothesis. According to this theory, based largely on the efficacy of drugs that act on this pathway of the central nervous system, an aberrant neurotransmission in certain brain circuits would be ultimately responsible for the appearance of psychotic symptoms. Roughly speaking, we could talk about an “excess” of dopamine in the mesolimbic pathways—responsible for the positive symptoms—and a “defect” of dopamine in the mesocortical pathways—responsible for the negative symptoms [3]. However, how does this aberrant neurotransmission in the brain of an adolescent or young adult occur?

### 1.1. Pathogenic Factors of Psychosis

Classically, researchers have focused on studying a number of pathogenic factors that could explain the etiology of psychotic spectrum disorders. For example, gene markers have been identified that may be related to the development of psychosis. Therefore, large-scale studies on the genomics of psychiatric diseases are beginning to illuminate genetic–molecular contributions to major mental disorders. On the one hand, we find common variants with a small effect size and, on the other, rare mutations with a larger effect size [4,5,6,7].

Another aspect long studied is childhood trauma. The literature indicate that the proportion of subjects with psychotic symptoms (especially hallucinations) who have suffered abuse in childhood is much higher than in the general population, indicating as possible bridges between both induction of social defeat and reduced self-value, sensitization of the mesolimbic dopamine system, changes in the stress and immune system, and concomitant changes in stress-related brain structures, such as the hippocampus and the amygdala [8,9].

However, if there is a risk factor long studied and related to the development of psychosis, it is undoubtedly the consumption of drugs of abuse [10,11]. For example, cannabis abuse induces transient psychotic disorders and is a predictor for an increased risk of schizophrenia (especially in young adults) [12]. Furthermore, it is associated with an earlier onset of psychosis, as well as transitioning to it in high-risk subjects [13]. Cocaine use has traditionally been associated with various psychiatric problems [14] and the appearance of psychotic symptoms is one of the most common complications [15]. Since cocaine has an agonist effect on dopaminergic neurotransmission at the central level, it is quite common for transient psychotic symptoms to occur in the context of acute intoxication [16]. On the other hand, in patients with primary psychotic disorders, such as schizophrenia, cocaine use is higher than in the general population [17], and its consumption has been associated with an earlier onset of psychotic illness [18]. Another psychostimulant with psychotic–mimetic properties similar to cocaine is methamphetamine (MA) [19]. Alcohol consumption and tobacco smoking have also been strongly associated with the presence of psychosis [20,21].

In analyzing the relationship between psychosis and drug abuse, a diagnostic and prognostic difficulty lies in the distinction between primary and substance-induced psychoses. Notably, due to the increasing use of drugs that possess the potential to produce psychosis, substance-induced psychotic symptoms are becoming a relatively frequent phenomenon. It has been reported that the onset of psychosis can take place up to 2 years earlier in substance abusers (and 2.7 years earlier if only cannabis is considered) [22]. Also, the propensity to develop psychosis seems to be related to the severity of use and dependence [11].

### 1.2. Psychosis, Drug Abuse and Neuroinflammation

Recently, the study of immuno-inflammatory factors that may be involved in the genesis of psychosis has become more important, describing a “pro-inflammatory” state enhanced in these disorders [23]. In relation to the inflammatory processes that may be underlying these alterations of the immune system, different hypotheses have been formulated: from the macrophage-T lymphocyte theory of Smith and Maes [24,25] to the theory of Schwarz et al. [26], who proposed the Th2 hypothesis. In turn, Monji et al. [27] described the microglia hypothesis: that activated central nervous system microglia release pro-inflammatory cytokines and free radicals, which induce abnormal neurogenesis that contributes to the pathophysiology of schizophrenia.

The study of the question is complicated, since drug abuse, in addition to being able to act as an independent risk factor for psychosis, has also been linked to immuno-inflammatory factors. Evidence suggests glial cells have an essential role in the development and maintenance of drug abuse by influencing neuronal and synaptic functions in multifaceted ways. For example, microglia and astrocytes perform critical functions in synapse formation and refinement in the developing brain [28,29].

Could substance use disorders and psychosis have a common origin? Are there biomarkers that can differentiate primary psychoses from induced psychoses? Below, we will discuss a number of inflammatory markers that have been linked to the onset of schizophrenia and other related diseases, and drug abuse. Our aim was to test the hypothesis that there could be an overlap between the different biomarkers found in psychosis disorders and substance use disorders, which would be able to tell us about a common pathway.

Subsequently, we try to relate these findings and ask ourselves a series of questions that can have a clinical impact on the care of our patients.

## 2. Methods

We carried out a narrative review of the literature through a search of studies in PubMed. We entered as search terms “psychotic disorders”, “schizophrenia”, “substance use disorder”, “cytokines”, “chemokines”, “endocannabinoid system”, “eicosanoids”, “lysophoshpolipids” and “bacterial products”. We filtered those studies that, from a narrative and nonsystematic approach, seemed most relevant to meet the aims of our work. We focused on clinical studies with human participants, although sometimes we cite findings from preclinical studies, especially when data in patients are scarce. At the same time, we did not consider for review those studies that only provided redundant information (for example, original research that was contained in already selected reviews).

## 3. Results

We summarized the main features and aims of the studies reviewed in Table 1 and Table 2. We integrated the results of the studies and present these in the following subsections.

### 3.1. Inflammation and Psychosis

Below we discuss several immuno-inflammatory markers that have been linked to psychosis spectrum disorders. We have summarized the data in Table 3.

#### 3.1.1. Cytokines

Cytokines are key signaling molecules of the immune system that exert their effects on the periphery and in the brain. They are produced by immune and non-immune cells and exert their effects by binding to their specific receptors present on a variety of target cells [90].

Cytokines (and cytokine receptors or antagonists) are key regulators of inflammation, which includes the activation and recruitment of immune cells and increases blood flow and vascular permeability. They coordinate both the natural and adaptive type of the immune system. They can be classified in several ways, the most used method being through their biological functions [91]. We can distinguish pro-inflammatory, anti-inflammatory and hematopoietic cytokines. The first group would include molecules such as IL-1, IL-6 and TNF; the second, IL-4, IL-10 and IL-13; and the third, IL-3 and IL-5.

It has been shown that in individuals with a first episode of psychosis there is an imbalance in favor of cytokines and other pro-inflammatory mediators compared to other anti-inflammatory mediators, suggesting that immune system abnormalities could be an endophenotype of the disease [30]. These findings suggest the possibility that cytokines (and cytokine receptor) and other inflammatory mediators may be a biomarker of disease recurrence and/or response to adjuvant anti-inflammatory therapy for a subset of patients. The interest in the question has led to attempts to replicate these findings, with disparate results, in individuals in a clinical phase prior to the first episode of psychosis itself—the so-called high clinical risk of psychosis—so that an adequate early identification could be associated with an early approach and better results [31,32].

What is the neurobiological process underlying the inflammatory insult that we can observe through the presence of pro-inflammatory cytokines, and that can trigger psychosis? On the one hand, an overactivation of astrocytes and microglia has been described, as well as presynaptic stimulation of dopaminergic receptors in the midbrain [92]. In addition, these pro-inflammatory cytokines affect the regulation of the kynurenine pathway and alter glutamatergic transmission [93].

Given the accumulation of evidence that the alteration in the cytokine count correlates with the existence of psychotic disorders and with their own symptoms, the question that researchers are asking is whether a laboratory analysis could serve as a reliable biomarker for the diagnosis of these diseases [94]. In a recent review, Dawidowski et al. [33] draw the following conclusions in this regard:-The elevation of the pro-inflammatory cytokine IL-6 is a finding that, with greater consistency between the different studies, could be considered a trait marker in schizophrenia; that is, one that is associated with hereditary and neurodevelopmental factors and, therefore, of susceptibility to the disease, remaining more or less stable during the different phases of it. Other potential trait markers would be elevations of the pro-inflammatory cytokines IL-1β and TNF-α, but there is insufficient evidence of their rise in high- or ultra-high-risk states of psychosis.-Elevation of the pro-inflammatory cytokines IFN-γ and TGF-β could be considered state markers of schizophrenia, i.e., those that are associated with the disease itself and its symptoms, and may vary depending on the phase of the disease or antipsychotic treatment. Other potential state markers are IL-4 and IL-10, although in this case, only in chronic schizophrenia and not in the first episodes of psychosis.

Alterations in cytokine levels have also been studied for other psychotic spectrum disorders that share clinical features with schizophrenia, such as bipolar disorder [95,96]. An interesting fact is that an overlap of elevated levels of the same peripheral cytokines has been observed in patients with both the first episode of schizophrenia and bipolar disorder with psychotic characteristics [34].

#### 3.1.2. Chemokines

Chemokines are small chemoattractant proteins (8–14 kDa) that have a role in mobilizing white blood cells to the site of inflammation at the peripheral level. In turn, they are involved in the development, maturation, survival and regeneration of nerve cells (glia and neurons) in the CNS. Chemokines have been classified into four subfamilies based on the number of N-terminal cysteine present: CXC, CC, XC and CX3C. However, functionally, the classic distinction between homeostatic and inflammatory chemokines is not as clear and many of the chemokines show a dual profile depending on the cell type and/or the inflammatory process triggered [97].

Chemokines play an important role in our biological systems and have been implicated in various diseases, including cancer, HIV/AIDS and atherosclerosis [98]. In general, chemokines trigger their functional activities by binding to G-protein-coupled receptors on the cell surface.

Thus, chemokines and their receptors play an important role in the nervous system, acting as trophic and protective factors that increase neuronal survival, regulate neuronal migration and synaptic transmission [35]. They are constantly secreted and are responsible for proper cell migration, for example, during the growth of the body. In the brain, the level of chemokines increases due to their secretion by many different cells: microglia, astrocytes, oligodendrocytes, and endothelial cells from blood vessels [99]. The blood–brain barrier plays a particular role, and its most important function is the precise exchange of chemical compounds between the CNS and the circulatory system. The integrity of the structure of the blood–brain barrier supports brain homeostasis and allows many neurological functions to be performed. Chemokines play a special role mainly in some CNS diseases, when damage to the blood–brain barrier and the blood–spine/brain fluid barrier causes leukocyte infiltration, triggering inflammatory processes [100].

Recently, chemokines have sparked the interest of numerous researchers regarding the role they may play in the genesis, development and natural history of psychotic disorders, especially schizophrenia. However, despite being considered a special type of cytokines, chemokines have so far not received the same attention as their older sisters, with the possible exception of IL-8 (CXCL8) [101,102,103,104]. However, accumulated evidence suggests that an increase in the serum level of pro-inflammatory chemokines (e.g., CCL2, CCL4, CCL11, CCL17, CCL22 and CCL24) strongly correlates with schizophrenic symptoms, including deficits in attention, working memory, episodic and semantic memory, and executive functions [35]. It has also been observed that fetal exposure to IL-8/CXCL8 may alter the early stages of neurodevelopment [36,37].

In the review we have discussed above, Dawidowski et al. [33] identify MCP-1 (CCL2) as another potential trait marker in schizophrenia, although there is a lack of solid data to support it in the population of first episode drug-naïve patients [105]. MCP-1 and IL-8 have also been linked to other psychiatric illnesses such as major depression or bipolar disorder [106,107,108], as well as postpartum psychosis [109].

#### 3.1.3. Endocannabinoids

The endocannabinoid system is defined as the set of two G protein-coupled receptors and seven transmembrane domains for Δ-9-tetrahydrocannabinol (THC): cannabinoid receptor type 1 (CB1R) and cannabinoid receptor type 2 (CB2R) [110]. CB1R is the prominent subtype in the central nervous system (CNS) and has attracted great attention as a possible therapeutic pathway in several pathological conditions, including neuropsychological disorders and neurodegenerative diseases. In addition, cannabinoids also modulate signal transduction pathways and exert profound effects on peripheral sites [111].

In 1992, the first endogenous ligand of the cannabinoid receptor type 1, a derivative of arachidonic acid that was called “arachidonylethanolamide” or “anandaminde”, was discovered. The fact that anandamide (AEA) cannot fully reproduce the effects induced by THC led to the discovery of another important endocannabinoid, 2-arachidonoylglycerol (2-AG) [112]. Endocannabinoids are prominently involved in suppressing synaptic transmission through multiple mechanisms, with microglial cells and astrocytes being shown to be capable of producing their own 2-AG or AEA [113].

Endocannabinoid-like compounds comprise several molecules that are structurally related to endocannabinoids although they do not bind to the same receptors. These endocannabinoid-like compounds include N-acylethanolamines and 2-monoacylglycerols. Their structural resemblance to the endocannabinoids makes them players in the endocannabinoid system, where they can interfere with the actions of the true endocannabinoids, because they, in several cases, engage the same synthesizing and degrading enzymes [114]. Although we could speak of an “endocannabinoidome”, for the purposes of review, we will simplify and include these compounds in the endocannabinoid system.

Since the endogenous cannabinoid system has been involved in different neuropsychiatric disorders, Minichino et al. [38] have wondered, in a recent review, whether such a system could be abnormal in people with psychosis. The main findings were increased levels of anandamide in CSF and blood and increased expression of CB1R in peripheral immune cells of people with psychotic disease, compared to healthy controls. In addition, the studies reviewed report that the severity of positive and negative symptoms was associated with a decrease in anandamide levels in the CSF [39,40], and with increased expression of CB1 and CB2 receptors in mononuclear cells in peripheral blood [41].

In summary, a higher endocannabinoid system tone is evidenced at an early stage of the disease and in individuals free of antipsychotics (naïve), as well as an inverse association with the severity of symptoms. Such an increase in tone was normalized after successful treatment. Since mechanically, the release of anandamide in the brain can provide retrograde inhibition of the mesolimbic hyperdopaminergic state [115] (and therefore a reduction of positive symptoms), a feedback mechanism has been proposed in which the CNS of a subject with psychosis “brakes” it through its endocannabinoid system, which would end up acting as a natural protector against the disease. Thus, successful treatments would decrease this need for negative feedback [116,117].

#### 3.1.4. Eicosanoids

Eicosanoids are oxidized derivatives of 20-carbon polyunsaturated fatty acids (PUFAs) formed by the cyclooxygenase (COX), lipoxygenase (LOX) and cytochrome P450 (cytP450) pathways. Arachidonic acid (ARA) is the usual substrate for eicosanoid synthesis. The COX pathways form prostaglandins (PGs) and thromboxanes (TXs), the LOX pathways form leukotrienes (LTs) and lipoxins (LXs), and the cytP450 pathways form various epoxy, hydroxy and dihydroxy derivatives [118].

These substances are highly bioactive, acting on many cell types through the G-protein-coupled receptors of the cell membrane, or as nuclear receptor ligands; and have been implicated in the pathophysiology of various diseases, such as cardiovascular disorders [119], gastrointestinal disorders, [120], cancer [121], Alzheimer’s disease [122] or even COVID-19 [123]. Many eicosanoids are known to have multiple effects, sometimes pleiotropic, on inflammation and immunity [124], and they control important cellular processes, including cell proliferation, apoptosis, metabolism and migration [125].

Eicosanoid biosynthesis is usually initiated by the activation of membrane phospholipase A2 (PLA2) in mammalian cells, which releases arachidonic acid, the main substrate of this process [126]. It has been observed that the activation of PLA2 modulates various neurotransmission systems that are involved in the pathophysiology of schizophrenia, such as the dopaminergic, serotonergic or glutamatergic systems [127]. A study by Smesny et al. [42] found an association between increased baseline PLA2 activity and a first episode of psychosis, as well as severity of schizophrenic symptoms. Previously, an increased cytoplasmic PLA2 activity in the serum of drug-free schizophrenic patients had already been reported in several studies [43,44,128].

A holistic explanation of the role that eicosanoid metabolism plays in the pathophysiology of psychosis would include increased activity of enzymatic pathways leading from lipid precursors in cell membranes to the synthesis of inflammatory products such as thromboxane A2 (TxA2), thromboxane B2 (TxB2) and prostaglandin E2 (PGE2) [129]. In a recent study, Pereira et al. [45] found an association between the elevation of the latter two and the existence of criteria for ultra-high risk of psychosis. These pathways could be slowed down by both antipsychotic drugs and drugs with direct anti-inflammatory action [129].

#### 3.1.5. Lysophospholipids

The second product of the enzymatic action of PLA2 are lysophospholipids which, in turn, are converted into platelet-activating factors. These lipid mediators play critical roles in the initiation, maintenance and modulation of neuroinflammation and oxidative stress [130]. The two main categories of lysophospholipids, characterized according to their backbone, are lysoglycerophospholipids and lysosphingolipids. In cells, these lysophospholipids are intermediate precursors for the biosynthesis of other lipids in cells. Their intracellular concentrations are lower than those found in extracellular fluids [131].

In recent years, some of these lysophospholipids were identified as signaling molecules [132]. To date, in vivo and in vitro studies have indicated that signaling by lysophosphatidic acid (LPA) and sphingosine 1-phosphate (S1P) (two of the best-studied lysophospholipids) may play multiple roles in the relevance of CNS disorders, based on model systems analyses for psychiatric disorders including schizophrenia, anxiety, memory impairment, and neurological disorders such as Alzheimer’s disease (AD) [133]. Thus, for example, the lysophosphatidic acid subtype LPA1 has been involved with neurochemical and biochemical changes and other schizophrenia-related alterations in null mutant rodents [134,135,136,137] and the downward regulation of the LPA1R gene in patients has been pointed out [46].

Autotaxin (ATX) is a secreted enzyme that produces LPA from lysophosphatidylcholine (LPC), a type of lysoglycerolphospholipid. A recent study in Japanese patients has revealed significantly lower levels of one of the species of LPA, LPA 22:6 (LPA—docosahexaenoic acid), in CSF of patients with schizophrenia and major depressive disorder compared to healthy controls [47]. In addition, it appears that there may be a negative correlation between plasma levels of LPA and score on the PANSS scale, which could be useful for symptomatic assessment in schizophrenia [48]. For their part, Madrid-Gambín et al. identified early markers of psychosis through a longitudinal study in which they analyzed the blood of 115 children (12 years of age) who were identified for the first time as patients with psychotic experiences at 18 years of age, evidencing an increase in the levels of 4 lysophosphatidylcholines [49].

All this informs us of an alteration in the metabolism of the LPC-ATX-LPA pathway that may already be present in stages much earlier than the clinical development of the disease, and that would play an important role in its genesis and subsequent development. However, the results are still very preliminary, and further study of these biochemical species is needed.

#### 3.1.6. Bacterial Products

For decades, the relationship that can exist between different infectious agents and the appearance of schizophrenia or schizophrenic spectrum disorders has been studied. For example, this relationship has been revealed in multiple ecological studies that relate the birth of subjects who will later develop psychosis in adulthood with influenza epidemics [138,139,140,141,142,143]. This relationship has also been studied for depressive disorder [144].

Sorensen et al. crossed data from the Copenhagen Perinatal Cohort and the Danish National Psychiatric Registry to test the hypothesis that a bacterial infection during pregnancy would confer an increased risk to offspring of developing schizophrenia in adulthood [145]. Exposure in the first trimester conferred an elevated risk of schizophrenia, which decreased from the second trimester. A relationship between the risk of schizophrenia and intrauterine exposure to rubella [146] and toxoplasma [147] has also been reported.

In recent years, these studies have shifted from eco-designs, which determine epidemic-based infection in populations, to research that has focused on reliable biomarkers in individual pregnancies [148]. On the other hand, there is a growing body of research that consistently involves the gut microbiome and the axis it forms with the CNS in the pathophysiology of various mental illnesses, including psychotic spectrum disorders [149].

Associations have been found between various bacterial products and psychosis. Lipopolysaccharides (LPS) are endotoxins found in the outer leaflet of the outer membrane of most Gram-negative bacteria [150]. Anti-LPS antibodies may serve as an indirect measure of LPS leakage into the circulation, which in turn would reflect a defect in intestinal barrier function [151]. Although, in a recent study, Delaney et al. [31] failed to find statistically significant differences in anti-LPS antibody levels between children, adolescents and young adults with psychosis, a population at ultra-high risk of psychosis, and healthy controls, they did find—for the psychosis group—a significant correlation between the increase in these antibodies and another well-established biomarker of psychosis, IL-6. This correlation could be due to the biological action produced by the binding of endotoxin to toll-like receptor 4 (TLR4), releasing the pro-inflammatory cytokine [152].

Maes et al. [50] have identified an overexpression of genes in patients with a first episode of psychosis involved in the response to LPS. These findings are in line with other studies reporting increased IgA/IgM responses to LPS from Gram-negative bacteria in this group of patients and deficit schizophrenia, and a significant association between these bacterial load indicators and general cognitive impairment and psychosis [51,52].

### 3.2. Inflammation and Drug Abuse

In Table 4 we have summarized the data presented on immuno-inflammatory markers in substance abuse disorders.

#### 3.2.1. Inflammatory Proteins and Drug Abuse

Cytokines and chemokines have emerged as a new study biomarker in substance use disorders, along with other immuno-inflammatory mediators [29,153]. This provides us a different framework for understanding the functional and behavioral changes that contribute to drug addiction [154].

Studies in consumer patients, as well as in animal models of addiction, also show an imbalance in favor of pro-inflammatory factors. On the one hand, it has been proposed that a pro-inflammatory state could alter the neuromodulation of circuits involved in the pathophysiology of addiction [154] and, on the other hand, that exposure to substances of abuse can modify the secretion of cytokines and chemokines [153].

The involvement of cytokines and chemokines in different substance use disorders related to psychosis has been reported. For example, serum/plasma and other biological samples elevations of IL-1ß [53], IL-8 [53], TGF-ß [54] and CCL11 [55], and a decrease in IL-2 levels [54], have been described for cannabis use. Conflicting results have been reported for IL-6 [53,56] and TNFα [53,56,57]. Other studies link lung inflammation to smoking marijuana use [155,156] and gene expression analyses performed on the epithelium of the airways of cannabis users show an upward regulation of several toll-like receptors involved in the initiation of the inflammatory cascade [157].

However, the relationship of cannabinoids to inflammation remains controversial and complex. There are reports that take into account the anti-inflammatory properties of these compounds through multiple mechanisms related to the immune system [158,159,160]. Overall, the accumulated evidence suggests that cannabis extracts could modulate such a biological system [153].

Araos et al. [58] have found plasma levels of TNF-α, IL-1β, CXCL12, CCL2 and CX3CL1 to be elevated during acute cocaine exposure in mice, but reduced in substance-dependent humans during periods of withdrawal. In addition, plasma levels of IL-1β, CXCL12 and CX3CL1 were positively associated with criteria of abuse and dependence.

Two clinical studies in cocaine users showed a decrease in IL-10 (anti-inflammatory) and IL-17 (pro-inflammatory) at the peripheral level in active and abstinence users, respectively [59,60]. Again, contradictory results have been reported for IL-6 [59,61,62,63]. A peripheral decrease in TNF-α, CCL2, CCL3 and TGF-α has also been reported in abstinent cocaine users, and a decrease in TGF-α levels has been proposed as a potential biomarker of dual diagnosis [60,62]. All these integrated findings would suggest an unbalanced immune system towards the pro-inflammatory pole during active consumption that would be inhibited in periods of withdrawal.

One chemokine receptor system involved in the pathophysiology of cocaine addiction is CCR5 [161], whose ligands are CCL3 and CCL5. It has also been noted that dysregulation of cytokine and chemokine signaling during cocaine exposure would perpetuate cocaine use disorder [162,163,164] and would play a prominent role in the psychiatric comorbidity present in chronic cocaine users [165].

Alcohol consumption has been linked to an altered neuroimmune response that includes the production and release of pro-inflammatory cytokines [166]. Cytokines and chemokines such as TNFα, IL-1α, IL-1ß, IL-6, IL-8, IL-12 and MCP-1, can be regulated by alcohol and alcohol-induced tissue damage [64]. A recent meta-analysis showed consistent evidence of IL-6 elevation in subjects diagnosed with alcohol use disorder compared to healthy subjects [65]. Recent studies further suggest that the transition from Th1 to Th2 cytokines contributes to liver fibrosis and cirrhosis, so ethanol disruption of cytokines and inflammation may contribute in multiple ways to a diversity of alcoholic pathologies [66].

García-Marchena et al. [67] found that plasma concentrations of IL-1ß, IL-6 and TNFα were increased, whereas plasma concentrations of IL-4, IL-17A and IFN-γ were decreased, in abstinent alcohol use disorder patients as compared with control subjects. Moreover, they found that changes in IL-6 and IL-17A plasma concentrations in alcohol use disorder patients were associated with the presence of liver and pancreatic diseases. It has also been reported that stress-related suppression of peripheral cytokines may predict future relapse in alcohol-dependent individuals [68].

Methamphetamine-mediated neuroinflammation has also been the subject of recent study, with stimulation in the secretion of cytokines and chemokines such as TNFα, IL-1ß, IL-6 and CCL2 having been described [69,70,71,72]. Tobacco smoking has been implicated in the production of many immune or inflammatory mediators, including both pro-inflammatory and anti-inflammatory cytokines [73]. However, the effects of tobacco on inflammation are complex, since it could produce, at the same time, a pro-inflammatory effect in certain places and anti-inflammatory in others, generating a transition from immune response Th1 to Th2 [74]. The use of other substances generally less involved in the pathophysiology of psychosis, such as opioids, has also been linked to immuno-inflammatory changes [167].

#### 3.2.2. Inflammatory Lipids and Drug Abuse

Small molecular weight lipids function as neuromodulators in the brain and, as such, play a role in the synaptic plasticity that occurs after exposure to drugs of abuse [168]. Recently, the important role played by the endocannabinoid system (ECS) in regulating the reinforcing and motivational properties of these substances has been described [169].

Exogenous administration of THC has been shown, for example, to influence endogenous cannabinoids circulating in humans, with altered levels of anandamide, 2-AG, palmitoylethanolamide and oleoylethanolamide [75]. In a recent study conducted on post-mortem brain tissue samples, Manza et al. [76] have reported that those subjects in whom physiologically restricted endogenous cannabinoid signaling may exist may be more vulnerable to the effects of chronic cannabis use on cortical thickness. Neuroimaging studies also show a decrease in the density of CB1 receptors in patients who use cannabis [77,78,79]. Genetic links to different critical elements of the ECS have also been proposed as a risk factor for cannabis dependence [169].

Two major human clinical studies analyzed concentrations of various substances in the endocannabinoid system in cocaine-using patients. Pavón et al. [80] carried out a study in which they characterized both free N-acyl-ethanolamines (NAEs) and 2-acyl-glycerols in abstinent cocaine addicts from outpatient treatment programs who were diagnosed with cocaine use disorder, and healthy control volunteers. While NAEs were found to increase, 2-acyl-glyceroles decreased in subjects with CUD compared to controls. Voegel et al. [81] compared hair concentrations of 2-arachidonylglycerol, anandamide (AEA), oleoylethanolamide (OEA) and palmitoylethanolamide (PEA) between recreational cocaine users (RCU), dependent cocaine users (DCU) and stimulant-naïve controls. Hair concentrations of OEA and PEA were significantly lower in DCU compared with RCU and controls. On the other hand, the interactions between the cannabinoid system and alcohol are well established. An increase in anandamide and 2-AG production has been reported in response to ethanol consumption [82]. García-Marchena et al. [83] have found that acylethanolamides were significantly increased in alcohol-dependent patients compared with control subjects.

Prostaglandin receptors are found in the cerebral cortex, hippocampus and midbrain [170]. The restoration of drug abuse is mainly mediated by the synergistic effect of the cannabinoid system and the products of arachidonic acid [171]. From this point of view, prostaglandins have an important role in drug addiction [172]. The influence of cocaine on the endothelial or plasma levels of PGI2, PDE2 or TXA2 has been reported [84,85,86]. In a review, Orio et al. [173] propose that, however, it is unknown whether the peripheral effects of cocaine on PG levels may influence central structures related to the addictive process. They relate, in different ways, the eicosanoids metabolism with cocaine addiction. Alterations in the production of prostaglandins, thromboxane and leukotrienes have also been reported in relation to alcohol consumption [87,88]. Some of these molecules have been linked to physical opioid dependence [174].

Because of its role in the control of both neural plasticity events and neurogenesis and its interrelation with the endocannabinoid system, LPA and its receptors might be involved in drug addiction-associated neuroadaptations [173]. Chronic cocaine administration induced conditioned locomotion deficits in LPA-1-receptor-null mice [175]. García-Marchena et al. have found a decrease in total LPA concentrations in patients with alcohol use disorder and healthy controls [89].

## 4. Discussion

Our review provides a panoramic—though not exhaustive—view of the main neuroinflammatory mechanisms that have been linked to psychotic disorders (especially schizophrenia) and substance use disorders. Although some prospective cohorts have been studied to analyze the risk of schizophrenia in adulthood of subjects who may have been exposed to toxic agents during development, most findings come from retrospective and observational studies. This should make us cautious with conclusions. However, some preclinical or in vitro data seem to support the findings in human participants [70,75,85,86]. Furthermore, data about inflammatory biomarkers in patients with dual diagnosis (psychosis and addiction) are reported in scientific literature and TGFα has been proposed as a potential biomarker of this double condition [60,62,80].

### 4.1. Drug Abuse and Psychosis: An Inflammatory Convergent Origin?

In this paper, we have reviewed, from a narrative approach, how both psychosis and drug abuse have been related to immuno-inflammatory changes and mechanisms. These would include a series of cellular and molecular systems in which an imbalance in favor of pro-inflammatory factors has been described, and there would also be an overlap between many signals found in both disorders. Considering that the reinforcing effects of drugs depend mainly on dopamine signaling in the nucleus accumbens [176], the main neuroanatomical substrate of many cardinal symptoms of psychosis, this overlap should not surprise us, and should make us wonder if both conditions could not ultimately have a common or convergent origin.

This new “inflammatory hypothesis” of mental disorders would not replace, but would complement, the classical hypotheses based on alterations in the metabolism and signaling of brain neurotransmitters. Many immune molecules interact with neurotransmitter systems and play an essential role in modulating synaptic function [154]. For example, TNFα modulates the traffic of AMPA-type glutamate receptors and GABA receptors [177,178,179], IL-1β modulates long-term potentiation [180,181], CCL2 and CXCL-12 regulate the release of glutamate, GABA and dopamine [182,183]. We further know that many of these neuroimmune molecules are also involved in all stages of neurodevelopment, play a key role in neurogenesis and gliogenesis, and interfere with neuronal migration and neurocircuit formation [184,185,186].

The role that chronic stress can have in psychosis spectrum disorders has long been discussed in the scientific literature [187]. In a recent review, Vargas et al. [188] list several converging mechanisms that would be involved in the development of psychosis. These would include hypothalamic–pituitary–adrenal axis activation, dopamine and glutamate dysregulation… and chronic inflammation. Therefore, peripheral inflammation resulting from stress exposure could lead to neuroinflammatory responses impacting cortico-amygdala function, as well as prefrontal cortex structure, function and development [189].

Considering all these findings, Millan et al. [190] have recently described an anomalous neurodevelopmental model for schizophrenia that we find especially interesting. Thus, in a developing individual, a first wave of biological and psychological insults would act (which would include factors such as prenatal infection, childhood trauma, malnutrition, etc.). Later, in a second wave, in the period comprising adolescence and early youth, drug use or social isolation would appear. All this would configure an abnormally developed brain and CNS that would make psychosis hatch in its positive dimension once the damage is produced.

In all this, it is worth asking: why do patients with psychosis have such a high rate of substance abuse? In general, the association between a substance use disorder and another mental illness is known as “dual diagnosis” [191]. Different etiological hypotheses have been proposed to explain this phenomenon [192]: that addiction is a consequence of primary mental illness (as a mechanism, for example, of “self-medication” in the face of psychological distress that these subjects would experience), that mental illness is a direct consequence of the addictive substance (as in induced psychoses) or, finally, that both disorders are independent and share common etiological factors. Within these, neuroinflammation could have a prominent role.

The search for that common inflammatory origin for psychosis and substance use disorders has led us to explore a neuroinflammatory and neurotoxic pathway that has its onset in the activation of the transcriptional factor NFκB. This is a ubiquitous transcription factor well known for its role in the innate immune response whose involvement in drug abuse has been the subject of a recent review [193]. Under normal conditions, NFκB is bound in the cytosol to its inhibitory complex, IκB. This prevents it from being transferred to the core and activating a pro-inflammatory cascade. However, certain stimuli, such as the activation of cytokine receptors or toll-like receptors (associated with bacterial infections), can cause the activation of an intracellular signaling pathway that leads to the phosphorylation of IκB, which would decouple from NFκB and allow the translocation of the latter to the cell nucleus [194]. There, they bind to specific DNA sequences in the promoter region of a wide range of genes encoding a number of inflammatory mediators [195,196]. All of this would generate a positive feedback loop [193].

García-Bueno et al. [30] have found increased NFκB activity in nuclear extracts of peripheral blood mononuclear cells (PBMC) from FEP patients versus healthy controls. They have proposed an integrative inflammatory hypothesis of schizophrenia that starts from the degradation of the inhibitory complex IκB and the activation by NFκB activity of pro-inflammatory enzymatic pathways through inducible nitric oxide synthase (iNOS) and isoform 2 of the enzyme cyclooxygenase-2 (COX-2). This would result in an accumulation of oxidative and nitrosative mediators, such as nitric oxide, peroxynitrite anion and PGE2, that would cause the depletion of endogenous antioxidant defenses and the attack on membrane phospholipids, causing cell damage through a lipid peroxidation process [197].

It has been shown, on the other hand, that NFκB activity can increase through diverse environmental stimuli that would include substances of abuse such as cocaine [198,199], opiates [200] or alcohol [201,202,203]. In addition, NFκB could play an important role in the functional regulation of mechanisms associated with the development of substance dependence. Thus, for example, in preclinical models it has been possible to reduce the sensitivity to the rewarding properties of cocaine by eliminating NFκB activity [199]. Finally, NFκB could influence drug seeking through a predominant biological role in the response to emotional stress or in physiological behavioral processes involved in memory formation [193].

NFκB would have an interesting counterpart that would act as a transcription factor dependent on ligands, and that would generate effects contrary to the receptor activated by the peroxisome proliferator, especially in its gamma isoform (PPARγ) [204]. Once PPARγ activated, it would translocate to the nucleus and modify the cell’s gene expression by inhibiting the inflammatory response. Interestingly, a known ligand of PPARγ is 15d-prostaglandin J2 (15d PGJ_2_), which, in turn, is a product of the COX pathway. Therefore, this pathway would be activated as a compensatory mechanism after the cascade initiated by NFκB, and the balance or imbalance of both would condition, respectively, the physiological or pathological result of neuroinflammatory processes. Recently, PPAR have been connected to the endocannabinoid system. These nuclear receptors, indeed, mediate the effects of anandamide and endocannabinoid-like substances such as OEA and PEA [205].

Given all these findings, we dare to hypothesize a model of neuroinflammation that can somehow link psychosis with substance abuse and that could be detected through various biological “signals” in blood and other tissue samples. Thus, in the first place, at a very early stage of development, a toxic environmental stimulus (for example, a bacterial infection) would provoke, through the stimulation of toll-like receptors and the participation of other inflammatory mediators, an activation of the master regulator of inflammation NFκB by degrading its inhibitory complex. This would activate the production of inflammatory mediators, among which would be lipids, such as eicosanoids or lysophospholipids, and proteins, such as cytokines and chemokines, which, in turn, would generate, by activating microglia, a “pro-inflammatory state” in the CNS of the young individual. It would trigger, concomitantly, and through the PPARγ pathway, an overactivation of natural anti-inflammatory systems, such as the endocannabinoid system, that, despite everything, could not counteract the previous effect. This pro-inflammatory state would influence the genesis of neurocircuits that would be pathologically formed, with alterations in critical pathways related to the search for pleasure or reward in various neuroanatomical substrates such as the nucleus accumbens. All this would generate a vulnerability to the development of psychosis or a substance use disorder. However, a second environmental insult is still necessary, in line with what was proposed by Millan et al., which would occur in adolescence with the effective consumption of a drug or another type of biological aggression to this vulnerable CNS. Finally, these pathological neurocircuits create an aberrant dopaminergic transmission in various parts of the brain, which would produce the known positive and negative symptoms of psychosis (and condition the development of dependence on the drug consumed).

In Figure 1, we have summarized our neuroinflammatory hypothesis about the common origin of psychosis and drug abuse.

### 4.2. Future Perspectives

If we look at the neurodevelopmental model of psychosis (which, as we have seen, we can also apply to substance use disorders to a large extent), we are faced with a bleak clinical picture: we are acting only at the end of the disease process, when brain damage is already established. In addition, we are acting not only at the end of the process at the chronological level, but at the tip of the iceberg of the biological processes underlying the etiopathogenesis of these disorders, which would be the alteration in the transmission of dopamine. This is the classic known mechanism of action of antipsychotic drugs. Notably, these drugs, with respect to which we have few reliable biomarkers to assess effectiveness or adherence (beyond the measurement of their plasma levels), are usually used indiscriminately and empirically without attending to the possible endophenotypes of disease that we may find. The new explanatory approaches to psychosis (whether primary, dual or toxic) should also help us to adopt novel clinical perspectives which change the way we understand its nature and, therefore, that suppose a change in the therapeutic approach to our patients.

A first question that could be addressed in light of these new findings is patient stratification. Recently, more and more authors dare to doubt that some diseases that we thought well categorized, such as schizophrenia, are not actually a construct whose validity is questionable [206]. Endophenotypes—measurable components not seen with the naked eye along the pathway between the disease and the distal genotype [207]—could serve as biological markers in schizophrenia. Some of these, related to microvascular abnormality [208] or impaired prostaglandin signaling [209], have been proposed. We propose, like other authors, to conceptualize psychosis as a “fever” of serious mental illnesses, a serious but non-specific indicator [210]. Having reliable biomarkers, such as inflammatory signals that we have analyzed in this review, can make us change the classic nomenclature still in force in international classifications and better distinguish some patients from others.

When we also introduce into this equation the variable associated with substance use, things get complicated. It is common that in many Western countries there is a separate network of care for mental health and drug addiction, which generates difficulties in the care of patients with dual diagnosis, and a deficit of professionals who are experts in treating both types of pathologies [211]. Despite the very high co-prevalence between substance use disorders and serious mental illnesses, clinical guidelines for both tend to have limited considerations for coexisting disorders in diagnosis, treatment and management, as evidenced by a recent review [212]. This, in our opinion, ends up generating a vicious circle in the circuits of attention to dual patients: they are continually referred from one care service to another without being treated properly. This is what has come to be called the “wrong door syndrome” [213].

In this review, we believe we have shown that psychosis and drug abuse are not only related from the epidemiological point of view, but also share a common vulnerability model and convergent etiopathogenic pathways. Therefore, their treatment must be integrated. Much importance has generally been granted to the distinction between “primary” and “induced” psychoses [214], despite the fact that a high rate of patients originally diagnosed with induced psychosis end up developing schizophrenia or bipolar disorder [215].

Another important perspective to consider in our review is response to treatment. There is still a high percentage of patients who are refractory to usual treatment, estimated at around 20–30% of the total [216]. However, if we consider as a therapeutic objective not only the remission of positive and/or negative symptoms but the functional recovery, then only one in seven patients will end up achieving it [217]. However, why do some patients respond well (or acceptably well) to conventional antipsychotics and others do not? A recent systematic review looking at multiple patient characteristics found that only a higher positive symptom score at baseline predicted a higher response rate [218]. We can say that, in a certain sense, we are quite blind, both when we start an antipsychotic treatment in a patient with psychosis and when we predict the response to second-line treatments.

However, something may be changing in recent times. There is already talk of a “precision” or “personalized” psychiatry that would consist of considering mental illness as a dysregulation, unique to each patient, of the molecular pathways that govern the development and functioning of the brain [219]. Within this approach, we think that an adequate phenotypic characterization of the inflammatory mechanisms involved in the genesis of psychosis can have important prognostic implications and, to some extent, predict the response to treatment.

In addition to predicting response to treatment, another important aspect of antipsychotic therapy is monitoring it. In our review, we have identified some immuno-inflammatory signals that could act as endogenous markers of response to psychotropic drugs. At least, we should consider the alterations in the concentrations of the pro-inflammatory cytokines IFN-γ and TGF-β as well as the variations we find in the lipid signaling of the molecules that are part of the endocannabinoid system, whose level of overactivation would also be mediated by the treatment. The final issue related to the treatment of people with psychosis that we want to discuss is the growing evidence of the effectiveness of some anti-inflammatory drugs, such as aspirin, estrogens, minocycline and NAC, for this condition [220].

Last but not least, we would like to point out some aspects that seem key to the prevention of psychosis. It may be the case that acting on adolescents in a clinical state of high risk for psychosis (which, until now, has been considered the priority objective of prevention programs) is already too late: a recently published review of Catalán et al. [221] concluded that there was insufficient evidence to recommend a specific treatment over others (including control conditions) to prevent the transition to psychosis. However, some interventions with potentially beneficial effects in patients at high risk for psychosis include omega-3 fatty acid supplementation (EPA and DHA), whose role in developmental psychopathology has been highlighted by Agostoni et al., and which appear to play a key role in resolving inflammation [222]. Furthermore, in this population, some neuroinflammatory signals could also act as early diagnostic or prognostic factors. Thus, a reliable trait marker that we could measure in this population and that seems to rise consistently during the prodromal phase of the disease is the cytokine IL-6. Eicosanoid products, such as thromboxane B2 (TxB2) and prostaglandin E2 (PGE2), could also be elevated at this phase. Finally, overactivation of the endocannabinoid system, or an alteration in the metabolism of the LPC-ATX-LPA pathway, may also be an early marker of disease, although further studies are required.

The fact that the care of adolescents in a high-risk mental state has not provided all the results we expected has led authors to propose that we move to a scenario of universal primary prevention [223]. Considering how brain inflammation, from the early stages of development, can condition the subsequent development of psychosis should lead us to improve pre- and perinatal care (especially in mothers with schizophrenia) [224,225], decrease prenatal infections and nutritional deficiencies [226], act in children at risk of psychosis by virtue of an increased familial or genetic risk [227], etc. Special mention deserves the action on drug abuse. Especially striking is the case of cannabis, having been calculated that, if high-potency cannabis were not available in Europe, 12 to 50% of the first episodes of psychosis could be prevented, depending on the geographical area [228]. We believe that, just as the fight against lung cancer has justified the proliferation of preventive campaigns to prevent nicotine abuse, a clear, forceful and clear response from the authorities cannot be delayed any longer: the use of cannabis considerably increases the risk of psychosis.

### 4.3. Limitations and Strengths

Of course, the review we have carried out has some limitations. First, the lack of a systematic search for results. However, it is known that sometimes narrative reviews like ours can offer some advantages and allow a more flexible and holistic approach to the question analyzed [229]. Second, we have not reviewed in-depth preclinical studies that could support the findings discussed in human participants. Finally, we must not forget an aspect that is not insignificant: to explore whether the association between inflammation and psychosis (and dual diagnosis) represents causality or rather a confounded epiphenomenon. Köhler-Forsberg et al. posed this question in a recent editorial [230]. In this sense, other mental disorders of high prevalence, such as major depression, have also been linked to neuroinflammatory mechanisms, including NFκB signaling [231]. However, the recurrent genetic overlap between different mental illnesses is already known, which should lead us to consider the sometimes-arbitrary limits between the diagnostic categories currently established in psychiatry [232]. Therefore, perhaps the common origin proposed in our review for psychosis and drug addiction is only a part of the convergent pathological pathways between numerous psychiatric disorders.

## 5. Conclusions

A significant number of neuroinflammatory signals that would include cytokines, chemokines, endocananbinoids, eicosanoids, lysophospholipids and bacterial products have been linked to psychotic disorders. Many of these markers also appear in substance use disorders, with an overlap between the two diagnostic categories.

Psychosis and substance use disorders, whose relationship has been studied for decades, may share a common origin. This would be an abnormal neurodevelopment where a first toxic stimulus would trigger an inflammatory imbalance in the body, with a production of pro-inflammatory lipids and proteins and an inability of endogenous anti-inflammatory systems, such as the endocannabinoid system, to counteract it. A possible convergent pathway is that which interrelates the transcriptional factors NFκB and PPARγ. Subsequently, a second toxic stimulus would precipitate the appearance of symptoms.

This finding may have implications for the prevention, diagnosis and treatment of our patients. However, we must be cautious with the interpretation of these findings, and more studies are needed in the future to delve into many of these issues.

## Figures and Tables

**Figure 1 biomedicines-11-00454-f001:**
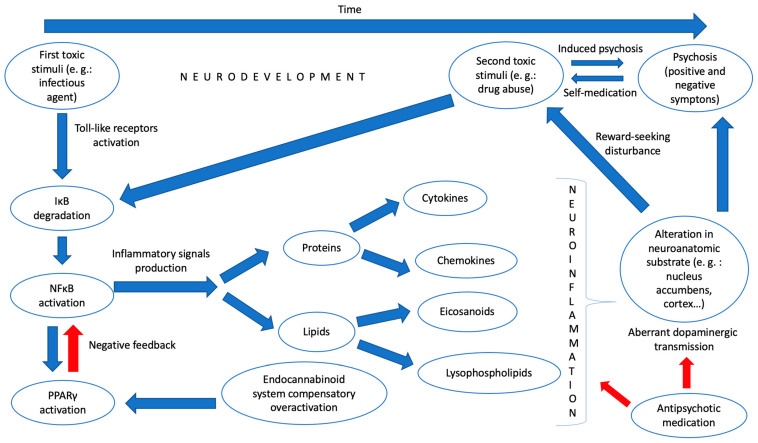
For psychosis to occur, at least two toxic environmental stimuli are necessary. Between them, an anomalous neurodevelopment is generated that confers a shared vulnerability to both drug abuse and psychotic disorders. One of the causes of this abnormal neurodevelopment is a neuroinflammatory state that has its origin in the degradation of the inhibitor complex κB (IκB), which causes the translocation to the cell nucleus of nuclear factor κB (NFκB) and the activation of a series of inflammatory signals (cytokines, chemokines, eicosanoids and lysophospholipids). There would also be an overactivation of the endocannabinoid system that would fail in its attempt to curb this pro-inflammatory state. Finally, an alteration in certain neuroanatomical substrates (nucleus accumbens, cerebral cortex…) would end up generating an aberrant dopaminergic transmission and would ultimately be responsible for both the reward-seeking disturbance observed in substance use disorders and the positive and negative symptoms of psychosis. Antipsychotic drugs would act by modifying this dopaminergic transmission and by slowing down neuroinflammation. In addition, drug abuse and psychosis would also be interrelated through the hypotheses of self-medication and the possibility of the appearance of induced disorders. The red arrows indicate a brake on the next step.

**Table 1 biomedicines-11-00454-t001:** Reviewed studies in psychotic disorders.

Study	Type	Population	Aim
García-Bueno et al. [30]	Observational original research	117 patients (mean age 23.91 ± 5.83 years; 69.2% male) during the first year after their first episode of psychosis (FEP) according to the DSM-IV criteria and 106 matched controls	To contrast the hypothesis that the physiological balance between interrelated pro-inflammatory/anti-inflammatory pathways may be disrupted in FEP
Delaney et al. [31]	Observational original research	47 children, adolescents and young adults with psychosis (mean age 24.30 ± 5.91; 63.83% male), 17 individuals at clinical high risk for psychosis (mean age 22.76 ± 3.68; 64.71% male) and 33 unaffected comparison-matched controls	To compare the levels of vitamin D, C-reactive protein (CRP), antibodies to lipopolysaccharide (LPS) and IL-6 between the groups
Park et al. [32]	Systematic review	8 studies of subjects at high-risk for psychosis (mean age range from 16.1 to 26.2; 31–100% male) versus matched controls (n range for cases from 14 to 76, and for controls from 39 to 115); 4 studies of high-risk converters versus non-converters (n range for converters from 14 to 56, and for non-converters from 60 to 129; mean age range from 16.1 to 21.9; 31–69% male)	To perform a meta-analysis of cytokine and C-reactive protein levels in high-risk psychosis
Dawidowski et al. [33]	Review	N/A	To present the most valuable evidence on cytokine dysregulation in schizophrenia, the links between cytokine levels and psychopathological presentation, as well as their alterations in response to antipsychotics
Lesh et al. [34]	Observational original research	69 first-episode schizophrenia-spectrum patients (mean age 19.9 ± 3.5; 88% male), 16 first-episode bipolar patients with psychotic features (mean age 21.4 ± 3.4 years; 75% male) and 53 healthy matched controls	To investigate differences in cytokine levels in plasma between individuals with first-episode schizophrenia, first-episode bipolar disorder with psychotic features, and healthy controls
Boczek et al. [35]	Review	N/A	To provide a summary of GPCR (G protein-coupled receptor)-acting neurotransmitters and chemokines and their role in schizophrenia
Ellman et al. [36]	Observational original research	17 cases diagnosed with schizophrenia (mean age 39.96 ± 1.78 years; 70.59% male) and 8 healthy matched controls	To determine the association between fetal exposure to IL-8 and structural brain changes among schizophrenia cases and controls
Brown et al. [37]	Review	N/A	To discuss the contribution of exposure to infection to the etiology of schizophrenia
Michino et al. [38]	Systematic review	18 studies in subjects with schizophrenia-related illnesses (44.3–75% male) versus healthy matched controls (n range for cases from 10 to 162, and for controls from 11 to 94).	To conduct a systematic review and meta-analysis of the blood and cerebrospinal fluid (CSF) measures of the endocannabinoid system (ECS) in psychotic disorders
Leweke et al. [39]	Observational original research	25 first-episode, antipsychotic-naïve schizophrenics with low frequency cannabis use (mean age 28.4 ± 8.8 years; 64% male); 19 first-episode, antipsychotic-naïve schizophrenics with high frequency of cannabis use (mean age 30.3; 84.21% male); and 81 healthy matched controls	To examine how cannabis use alters levels of anandamide in cerebrospinal fluid (CSF) in schizophrenic patients
Giuffrida et al. [40]	Observational original research	47 antipsychotic-naïve first-episode paranoid schizophrenics (mean age 28.9 ± 9 years; 53.6% male), 13 dementia patients (mean age 77.8 ± 7.8 years; 46.2% male), 22 affective disorder patients (mean age 44.7 ± 15.8 years; 46.2% male), 71 schizophrenics patients treated with antipsychotics (mean age 29.1 ± 10.1 years; 80.3% male), and 84 volunteers with no family history of psychiatric disturbances (mean age 27.9 ± 2.8 years; 53.6% male)	To examine the role of endocannabinoid signaling in psychotic states by measuring levels of the endocannabinoid anandamide in cerebrospinal fluid (CSF) of acute paranoid-type schizophrenic patients
Chase et al. [41]	Observational original research	35 participants with schizophrenia (mean age 36.14 ± 11.64 years; 51.43% male) and 35 healthy matched controls	To measure mRNA levels of cannabinoid receptors (CBRs) in human peripheral blood mononuclear cells (PBMCs)
Smesny et al. [42]	Observational original research	48 patients with first episode of schizophrenia (mean age 19.41 ± 2.94 years; 72.92% male) and 22 healthy matched controls	To investigate if intracellular phospholipases A2 (inPLA2) activity is associated with symptoms severity and treatment response in first-episode schizophrenia (FES)
Noponen et al. [43]	Observational original research	39 schizophrenics patients (mean age 44.8 ± 10.9 years; 100% male), 26 psychiatric non-schizophrenics patients (mean age 47.2 ± 12.6 years; 100% male) and 26 non-psychiatric and healthy volunteers (mean age 38.3 ± 11 years; 53.85% male)	To measure serum phospholipase A2 (PLA2) activity in all participants
Gattaz et al. [44]	Observational original research	20 paranoid schizophrenic patients (mean age 32 ± 12 years; 45% male), 6 non-schizophrenic psychiatric patients (mean age 37 ± 11 years; 0% male), and 21 non-psychiatric and healthy volunteers (mean age 31 ± 7 years; 47.62% male)	To investigate the activity of phospholipase A2 in the plasma of schizophrenic patients and healthy controls, as well as in a small group of non-schizophrenic psychiatric patients
Pereira et al. [45]	Observational original research	67 ultra-high-risk (UHR) for psychosis individuals (mean age 23.5 ± 3.4 years; 41.79% male) and 55 healthy matched controls	To investigate whether the study of the inflammatory COX-2 pathway through the quantification of the eicosanoid levels can be a useful approach for the characterization of ultra-high-risk (UHR) for psychosis individuals
Bowden et al. [46]	Observational original research	14 individuals with schizophrenia (mean age 35.86 years; 78.57% male) and 14 non-psychiatric matched controls	To generate gene expression profiles from peripheral blood lymphocytes from participants
Omori et al. [47]	Observational original research	27 patients with schizophrenia (mean age 40.1 ± 10.0 years; 51.9% male), 26 patients with major depressive disorder (mean age 40.4 ± 8.3 years; 50% male) and 27 healthy matched controls	To investigate the levels of lysophospholipid species, including LPA and related metabolic enzymes, in cerebrospinal fluid (CSF) of patients with major depressive disorder and schizophrenia
Gotoh et al. [48]	Observational original research	Study 1 (cerebrospinal fluid levels): 49 patients with schizophrenia (mean age 39.7 ± 10.5 years; 48.98% male) and 49 healthy matched controls; Study 2 (plasma levels): 42 patients with schizophrenia (mean age 40.7 ± 8.9 years; 54.76% male) and 44 healthy matched controls	To measure lysophosphatidic acid (LPA) levels by enzyme-linked immunosorbent assay in cerebrospinal fluid (CSF) (study 1) and plasma (study 2) samples
Madrid-Gambín et al. [49]	Observational original research	48 patients with psychotic experiences at 18 years of age but who did not have psychotic disorder (45.83% male) and 67 controls (58.2% male) of the same birth cohort	To identify early biomarkers of psychotic experiences in plasma samples of the participants when they were 12 years old
Maes et al. [50]	Secondary data analysis	Subjects with first episode of psychosis and schizophrenia	To delineate a) the characteristics of the protein–protein interaction (PPI) network of antipsychotic-naïve first-episode psychosis (AN-FEP) and its transition to schizophrenia; and b) the biological functions, pathways and molecular patterns which are over-represented in FEP/schizophrenia
Maes et al. [51]	Observational original research	21 subjects with first episode of schizophrenia (mean age 38.4 ± 12 years; 47.62% male), 58 subjects with multiple-episode schizophrenia (mean age 41.9 ± 10.7 years; 59.6% male), and 40 healthy matched controls	To delineate (a) the differences in several pathological and neuroimmune pathways between stable-phase, first- (FES) and multiple (MES)-episode schizophrenia and (b) the pathways that determine the behavioral–cognitive–physical–psychosocial (BCPS) deterioration in FES/MES
Maes et al. [52]	Observational original research	80 schizophrenia patients and 38 healthy controls (overall mean age 39.95 years; 55.93% male)	To measure plasma IgA/IgM responses to five Gram-negative bacteria in association with IgM responses to malondialdehyde (MDA) and azelaic acid in participants

**Table 2 biomedicines-11-00454-t002:** Reviewed studies in substance use disorders.

Study	Type	Population	Aim
Bayazit et al. [53]	Observational original research	34 patients with cannabis use disorder (mean age of 26 ± 9 years; 100% male) and 34 healthy matched controls	To evaluate oxidant and antioxidant status and cytokine levels in individuals with cannabis use disorder
Pacifi et al. [54]	Observational original research	37 polydrug consumers of 3,4-methylenedioxymethamphetamine (MDMA) and cannabis (mean age 23.6 ± 3.5 years; 51.4% male) compared to 23 cannabis users only (mean age 22 ± 1.9 years; 34.8% male) and 34 non-consumers (mean age 22 ± 2.6 years; 26.5% male)	To investigate cell-mediated immune function and the occurrence of mild infectious diseases in participants
Fernandez-Egea et al. [55]	Observational original research	18 current cannabis users (mean age 22.3 ± 5.1 years; 66% male), 33 past cannabis users (mean age 23.6 ± 4.3 years; 66% male) and 36 subjects who never used cannabis (mean age 24.3 ± 4.7 years; 68% male)	To explore the possibility that cannabis use influenced CCL11 chemokine plasma levels
Ribeiro et al. [56]	Observational original research	21 cannabis users (mean age 28.6 ± 8.54 years; 100% male), 12 cocaine users (mean age 37.8 ± 4.87 years; 100% male), 27 cannabis-plus-cocaine users (mean age 32.3 ± 7.91 years; 100% male) and 21 non-drug users (mean age 33.42 ± 9.73 years; 100% male)	To investigate the effects of illicit drugs on circulating lipopolysaccharide (LPS), systemic inflammation and oxidative stress markers in drug users
Keen et al. [57]	Observational original research	77 lifetime non-drug users (32% male), 46 lifetime marijuana only users (67% male) and 45 lifetime marijuana and other drug users (60% male); overall median age = 47 years	To explore potential differential effects of lifetime marijuana use on interleukin-1 alpha (IL-1α) and tumor necrosis factor (TNF) in a community-based sample
Araos et al. [58]	Observational original research	82 abstinent cocaine users who sought outpatient cocaine treatment (mean age 36.9 ± 7.8 years; 18.3% male) and 65 healthy matched controls	To examine the plasma pro-inflammatory cytokine and chemokine profile in participants
Moreira et al. [59]	Observational original research	12 cocaine users (mean age 24.92 ± 4.80 years; 75% male) and 24 healthy matched controls	To investigate serum levels of pro and anti-inflammatory cytokines, IL-6 and IL-10, respectively, in cocaine users from a young population-based sample
Maza-Quiroga et al. [60]	Observational original research	79 patients diagnosed with cocaine use disorder (CUD) in abstinence (34.87 ± 7.18 years) and 81 healthy matched controls	To test the hypothesis that patients with CUD in abstinence might have altered circulating levels of signaling proteins related to systemic inflammation
Halpern et al. [61]	Experimental original research	30 healthy participants with a history of cocaine use (16 females (mean age 25.8 ± 1.0 years)) and 14 males (mean age 27.8 ± 1.5 years))	To measure neuroendocrine and immunological responses to IV injection of 0.4 mg/kg cocaine or saline placebo
Pedraz et al. [62]	Observational original research	55 abstinent cocaine-addicted subjects diagnosed with lifetime cocaine use disorders (40 men (mean age 37.1 ± 6.7 years) and 15 women (mean age 42.8 ± 6.2 years)) and 73 healthy matched controls	To evaluate the sex differences in psychiatric comorbidity and the concentrations of plasma mediators that have been reported to be affected by cocaine
Levandowski et al. [63]	Observational original research	42 crack cocaine-dependent women (mean age 31.22 ± 7.83 years) and 52 healthy matched controls	To investigate the association between plasma interleukin 6 (IL-6) levels and executive function (EF) in crack cocaine-dependent women
Anchur et al. [64]	Review	N/A	To discuss cytokine biomarker candidates for alcohol abuse and alcoholism
Moura et al. [65]	Systematic review	23 studies of subjects with alcohol use disorder (56–100% male) and healthy matched controls (n range for cases from 9 to 42, and for controls from 6 to 46)	To assess differences in blood inflammatory cytokines between people with alcohol use disorder (AUD) and healthy controls (HC)
Crews et al. [66]	Review	Alcohol users	To discuss the contribution of cytokines in alcohol use and alcoholic pathologies
García-Marchena et al. [67]	Observational original research	85 abstinent subjects with alcohol use disorders (mean age 47.16 ± 7.27 years; 68.24% male) and 55 healthy matched controls	To explore possible associations in circulating plasma cytokine concentrations in abstinent patients diagnosed with alcohol use disorders
Fox et al. [68]	Experimental original research	33 alcohol-dependent individuals (12 with low depressive symptoms (mean age 38.7 ± 2.6 years; 86% male) and 21 with high depressive symptoms (mean age 39.1 ± 2.4 years; 84% male)) and 37 social drinkers (21 with low depressive symptoms (mean age 31.9 ± 2.2 years; 62% male) and 16 with high depressive symptoms (mean age 35.7 ± 2.4 years; 38% male))	To examine cytokine responses to stress in alcohol-dependent individuals and social drinkers, both with and without subclinical depression
Clark et al. [69]	Review	N/A	To review evidence that psychostimulants of abuse (cocaine, methamphetamine, ecstasy) are associated with activation of the innate immune response
Loftis et al. [70]	Observational original research (+preclinical experimental model)	20 adults in remission from methamphetamine (MA) dependence (mean age 33.3 ± 1.84 years; 70% male) and 20 healthy matched controls (+preclinical model) Preclinical model: 32 male C57BL/6J mice that were administered MA (1 mg/kg) or saline subcutaneously for 7 consecutive days	To test the hypothesis that immune factors, such as cytokines, chemokines and cellular adhesion molecules, contribute to MA-induced immune dysfunction, neuronal injury and persistent cognitive impairments
Yamamoto et al. [71]	Review	N/A	To discuss long-term decreases in markers of biogenic amine neurotransmission in methamphetamine and MDMA users
Yamamoto et al. [72]	Review	N/A	To highlight experimental evidence that methamphetamine and MDMA increase oxidative stress, produce mitochondrial dysfunction and increase inflammation, all of which converge and culminate in long-term toxicity to dopaminergic and serotonergic neurons
Qiu et al. [73]	Review	N/A	To review the influence of smoking on major components of both innate and adaptive immune cells, and summarize cellular and molecular mechanisms underlying effects of cigarette smoking on the immune system
Lee et al. [74]	Review	N/A	To discuss specific mechanisms by which cigarette smoking affects host immunity
Walter et al. [75]	Experimental original research (+preclinical experimental model)	15 healthy young men (mean age, 28.1 ± 3.1 years) and 15 women (mean age, 26.6 ± 2.4 years); preclinical model: 20 female Lewis rats (10 rats per treatment group) that were administered 3 mg/kg of the CB1/CB2-agonist WIN 55,212-2 dissolved in 1:1 dimethyl sulfoxide/phosphate buffer	To investigate effects in circulating concentrations of endocannabinoids after administration of a single oral dose of 20 mg delta9-tetrahydrocannabinol (THC) to 30 healthy volunteers and comparison with placebo
Manza et al. [76]	Observational original research	89 individuals with cannabis dependence (mean age 28.6 ± 3.5 years; 71.9% male) and 89 healthy matched controls	To lend insight into biological processes that might link chronic cannabis use to brain structural abnormalities
Spindle et al. [77]	Experimental original research	10 females with cannabis use disorder (mean age 23.2 ± 2.7 years), 10 female healthy controls (mean age 25.5 ± 5 years) and 7 male non-cannabis consumers (mean age 29.6 ± 6.9 years)	To explore the relation between acute cannabis effects and mood/craving/withdrawal and CB_1_ receptor availability
Ceccarini et al. [78]	Observational original research	10 chronic cannabis users (mean age 26.0 ± 4.1 years; 80% male) and 10 healthy matched controls	To test that chronic cannabis use may alter specific regional CB1 receptor expression
Hirvonen et al. [79]	Observational original research	30 male cannabis smokers (mean age 28 ± 8 years) and 28 healthy matched controls	To demonstrate downregulation of brain cannabinoid CB_1_ (cannabinoid receptor type 1) receptors after chronic exposure to cannabis in humans
Pavon et al. [80]	Observational original research	88 abstinent cocaine addicts (mean age 36.9 ± 8.4 years; 88.6% male) and 46 matched healthy control	To evaluate circulating endocannabinoid-related lipids as biomarkers of cocaine use disorder
Voegel et al. [81]	Observational original research	73 chronic cocaine users (mean age 31.3 ± 9.5 years; 67.12% male) and 67 healthy matched controls	To investigate alterations of hypothalamic–pituitary–adrenal (HPA) axis and endocannabinoid (eCB) system markers in individuals with chronic cocaine use disorder
Kunos et al. [82]	Review	N/A	To show evidence accumulated over the last two decades to indicate that both the addictive neural effects of ethanol and its organ toxic effects in the liver and elsewhere are mediated, to a large extent, by endocannabinoid signaling
García-Marchena et al. [83]	Observational original research	79 abstinent alcohol-dependent subjects (49.13 ± 9.6 years; 65.8% male) and 79 healthy matched controls	To characterize the plasma acylethanolamides in alcohol dependence
Yang et al. [84]	Review	N/A	To discuss the role of cyclooxygenase-2 (COX-2) in synaptic signaling
Copeland et al. [85]	Experimental preclinical study	Preclinical model: 51 young, male rabbits that were randomly selected to receive subcutaneous injections of either cocaine or saline solution	To study cocaine-induced alterations in cerebrovascular function and metabolism
Mastrogiannis et al. [86]	Experimental in vitro model	Endothelial cells were isolated from human umbilical veins derived from uncomplicated pregnancies with a negative history of drug abuse	To investigate the possible effects of cocaine on prostacyclin and prostaglandin (PG) E2 production from endothelial cells derived from human umbilical cord
Anton et al. [87]	Review	N/A	To review the nature and role of prostaglandins in the central nervous system, and what is known about the effect of ethanol on prostaglandin production in brain
Murphy et al. [88]	Review	N/A	To discuss effects of ethanol in synthesis of prostaglandins and leukotrienes
García-Marchena et al. [89]	Observational original research	55 abstinent alcohol use disorder (AUD) patients (mean age 47.7 ± 7.7 years; 81.8% male) and 34 healthy matched controls	To investigate whether the relevant species of LPA were associated with clinical features of alcohol addiction

**Table 3 biomedicines-11-00454-t003:** Immuno-inflammatory markers in psychotic spectrum disorders.

Type of Biomarker	Main Findings	References
Cytokines	↑ Pro-inflammatory cytokines ↓ Anti-inflammatory cytokines ↑ IL-6 as a trait marker of schizophrenia (lower level of evidence for IL-1ß and TNF-α) ↑ IFN-γ and TGF-β as state markers of schizophrenia (lower level of evidence for IL-4 and IL-10) Overlap between altered cytokine profiles in schizophrenia and bipolar disorder	[30,31,32,33,34]
Chemokines	Correlation between ↑ pro-inflammatory chemokines (e.g., CCL2, CCL4, CCL11, CCL17, CCL22, and CCL24) and schizophrenic symptoms Relationship between exposure to IL-8/CXCL8 and impaired neurodevelopment MCP-1 (CCL2) as a trait marker of schizophrenia	[33,35,36,37]
Endocannabinoids	↑ Anandamide in CSF and blood ↑ Expression CB1R in peripheral immune cells Negative correlation between positive and negative symptoms and anandamide levels in CSF Positive correlation between positive and negative symptoms of CB1R and CB2R in mononuclear cells	[38,39,40,41]
Eicosanoids	↑ Baseline PLA2 activity in FEP and drug-free schizophrenic patients Positive correlation between PLA2 activity and severity of schizophrenic symptoms ↑ PGE2 and TxB2 in patients with criteria for ultra-high risk of psychosis	[42,43,44,45]
Lysophospholipids	Downward regulation of the LPA1R gene ↓ LPA 22:6 in CSF Negative correlation between plasma levels of LPA and score on the PANSS scale ↑ Lysophosphatidylcholines years before the onset of the disease	[46,47,48,49]
Bacterial products	Positive correlation between ↑ anti-LPS antibody and IL-6 Overexpression of genes in patients with first episode of psychosis involved in the response to LPS ↑ Increased IgA/IgM responses to LPS in FEP and deficit schizophrenia and association between them and general cognitive impairment	[31,50,51,52]

CSF: cerebrospinal fluid; FEP: first episode of psychosis; PANSS: positive and negative syndrome scale. ↑: Elevation of the biomarker. ↓: Descent of the biomarker.

**Table 4 biomedicines-11-00454-t004:** Immuno-inflammatory markers in substance use disorders.

Type of Biomarker		Main Findings	References
Inflammatory proteins	Cannabis	↑ IL-1ß, IL-8, TGF-ß, CCL11 ↓ IL-2 Conflicts results for IL-6, TNFα	[53,54,55,56,57]
	Cocaine	↓ TNF-α, TGF-α, IL-1β, CXCL12, CCL2, CCL3 and CX3CL1 in substance-dependent humans during periods of withdrawal (but some of them elevated during acute cocaine exposure in mice) ↓ TGF-α as a potential biomarker of dual diagnosis Positive correlation between plasma levels of IL-1β, CXCL12 and CX3CL1, and criteria of abuse and dependence ↓ IL-10 in active users ↓ IL-17 in abstinence users Conflicts results for IL-6	[58,59,60,61,62,63]
	Alcohol	Dysregulation of TNFα, IL-1α, IL-1ß, IL-6, IL-8, IL-12 and MCP-1 ↑IL-1ß, IL-6 and TNFα in abstinent users ↓IL- 4, IL-17A and IFN- γ in abstinent users Transition from Th1 to Th2 cytokines Correlation in changes in IL-6 and IL-17A with liver and pancreatic diseases Stress-related suppression of peripheral cytokines may predict future relapse in alcohol-dependent individuals	[64,65,66,67,68]
	MA	↑ TNF α, IL-1ß, IL-6 and CCL2	[69,70,71,72]
	Tobacco	Dysregulation in production of pro/anti-inflammatory cytokines	[73,74]
Inflammatory lipids	Cannabis	Altered levels of anandamide, 2-AG, PEA and OEA ↓ Density CB1R in brain	[75,76,77,78,79]
	Cocaine	↑ NAEs in abstinent cocaine addicts ↓ 2-AG in abstinent cocaine addicts ↓ Hair concentrations of OEA and PEA in DCU compared with RCU and controls Altered plasma levels of PGI2, PDE2 and TXA2	[80,81,84,85,86]
	Alcohol	↑ Anandamide, 2-AG and NAEs Alterations in the production of prostaglandins, thromboxane and leukotrienes ↓ Total LPA	[82,83,87,88,89]

MA: methamphetamines; 2-AG: 2-acylglycerols; PEA: palmitoylethanolamide; OEA: oleoylethanolamide; NAEs: N-acyl-ethanolamines; DCU: dependent cocaine users; RCU: recreational cocaine users. ↑: Elevation of the biomarker. ↓: Descent of the biomarker.

## Data Availability

Not applicable.

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
