# Peer review of "The Inflammatory Signals Associated with Psychosis: Impact of Comorbid Drug Abuse"

_biomedicines, 2023, doi:10.3390/biomedicines11020454_

Round 1

Reviewer 1 Report

The idea discussed  by authors in this  review is very intriguing but to support the hypothesis reported some major points have to resolved:

1. Beyond the conclusions, the studies reported should be critically discussed. For each study should be reported:
For clinical studies: type of study, end point, criteria of inclusion,  mean age, sex etc.
For preclinical studies: experimental model, aim.
For addiction: substances involved,  age of addiction, comorbidities.
The data obtained in these studies were replicated by different authors?
2. The data about inflammatory biomarkers in patients with double diagnosis (psychosis and addiction) are reported in scientific literature?
3. The authors should discuss the preclinical (in vitro and in vivo) data to support the Clinical data reported.
4. The data in animal models of both psychosis and addiction are reported in scientific literature?
5. Cigarette smoking is strongly associated with psychosis. This should be included in the review. The same for alcohol disorders.
6. The stress stimuli are considered the trigger for a number of psychiatric disorders like psychosis and drug abuse. In the stress models a number of inflammatory biomarkers included NFkB and PPAR.  The authors should show that these biomarkers are altered in psychosis and drug addiction but not in other psychiatric disorders (i.e. depressive conditions).  Otherwise these biomarkers are common for stress stimuli hence for different psychiatric disorders.

Author Response

Dear reviewer,

First of all, thank you very much for your comments, which are undoubtedly aimed at improving the quality of our study. We have considered most of them.

Indeed, the narrative approach of our review led us to an integrated presentation of the results where the reviewed studies were not critically presented. We are going to add two tables in the manuscript with a brief description of all of them.

In our study, we have focused mainly on clinical studies in humans, although we have also made reference to some preclinical studies. We have added a "Methods" section where we explain this in greater depth and the lack of a comprehensive review of preclinical studies has been noted in a limitations section. However, at the beginning of the discussion we have referenced studies that support the conclusions obtained both in preclinical models and in patients with dual diagnosis.
We have also included in the review studies on alcohol and tobacco as very appropriately you pointed out.

Finally, we also discuss the possible role that neuroinflammatory pathways may have in other psychiatric illnesses such as major depression. The recurrent genetic overlap between different mental illnesses is already known, so perhaps the common origin proposed in our review for psychosis and drug addiction is only a part of the convergent pathological pathways between numerous psychiatric disorders.

We are applying all this changes in the revised version of the manuscript and we will be able to resubmit it tomorrow.

King regards,

Authors

Reviewer 2 Report

It is a very interesting manuscript, full of novel hypotheses and well-explained concepts. Through a narrative review methodology, the authors expose the main neuroinflammatory markers common to psychosis and substance abuse, trying to develop a biological model based on an hypothetical common origin. The review is complete (not exhaustive) and, above all, didactic. The tables and figure 1 stand out. I consider that their publication may be of interest to readers, both in the field of psychiatry and neuroimmunology.

I add some comments, in case they help:

- The review is quite long, compared to similar scientific publications in high-impact journals. I would recommend shortening it slightly, trying to focus on the main hypothesis. Some paragraphs about what psychosis is and why it occurs (in Introduction), or the development of therapeutic and preventive implications (in Discussion) could be shortened.

- I would remove the vague last sentence of the summary ("this may have implications for the prevention, diagnosis and treatment of our patients"), replacing it with a more concrete conclusion.

- in table 1 there seems to be a mistake (91.92 in the first column)

- finally, a more relevant topic. I believe that neuroinflammatory alterations associated with stress (in subjects without psychosis or substance abuse) should be also taken into account. Likewise, trauma is associated with an imbalance between pro and anti-inflammatories in the brain. For all these reasons, I miss the role of stress and trauma in the proposed model, and I think it can be incorporated relatively easily.

I would like to congratulate the authors for their expository and synthesis capacity.

Author Response

Dear reviewer,

Thank you very much for your comments aimed at improving the quality of our study. We have considered most of them.

As you say, perhaps our manuscript has gone too long in some respects. For this reason we have made an effort of synthesis summarizing many ideas in the discussion.
However, other reviewers have asked us to extend some aspects of the introduction, change the presentation of the results, or deepen the critical discussion of the reviewed studies. Therefore, some aspects have had to be summarized and others extended; but we hope that in the end the result will be a work more focused on the fundamentals.

We have removed the last statement from the abstract as you recommended and corrected other minor errors detected. We have also included the role of stress and trauma in both the introduction and discussion.

We are applying all this changes in the revised version of the manuscript and we will be able to resubmit it tomorrow.

King regards,

Authors

Reviewer 3 Report

This is a very interesting narrative review aiming to investigated the association of inflammatory markers in patients with psychosis with/without comorbid substance use disorders. The paper if well-written,however, several changes are recommended before considering it for publication.

Abstract.

1-I prefer to use others terms than "pathological entities".The last sentence of the background is not necessary.

2-The methods subsection of the abstract should better describe the inclusion and exclusion criteria. I recommend to start: "A narrative review as carried out...

3-The results subsection is really short, and the conclusions are long. I recommend to expand the results section and to summarize the conclusions.

Introduction

1-The introduction section is really brief. I recommend to expand the inflammatory hypotheses of psychosis, particularly, in the context of schizophrenia, and inflammatory markers in substance use disorders. At the end of the introduction, the authors should explain the main aims of this narrative review.

Methods.

1-The authors should describe how did the narrative review. Which electronic searches did they do? Search terms?

Results

The results section should be organized according to the findings. How many articles did the authors found? Although it is not a systematic review, a flow chart including the screening and selection process.

Is the section 3 summarizing the common inflammatory pathways between psychosis and substance use disorders? This should be clarified.

Discussion.

The discussion section is too long. I recommend to divide it into "Discussion", Limitations and strenghts,  Future perspectives.

The conclusions should be rephrased. They are more an opinion and summarize of results, than a conclusions section.

Author Response

Dear reviewer,

First of all, thank you very much for your comments, which are undoubtedly aimed at improving the quality of our study. We have considered most of them.

We have made all the changes that you propose in the abstract section.

We have expanded the introduction and further discussed the inflammatory hypothesis in psychosis and drug abuse. In this sense, the discussion on the epidemiological relationship between drug abuse and psychosis, which appeared later in the original version of the manuscript, is now included in the introduction, albeit summarized.

In addition, we have made a more didactic division of the manuscript into Introduction, Methods, Results and Discussion, in order to be able to apply the suggestions that you have correctly made to us. We have tried to summarize many aspects of the discussion and focus on the essentials. We have made the division that you propose including the sections "Limitations and strengths" and "Future perspectives".

We have also changed the wording of some conclusions so that they can be more objective, keeping in mind that this is a narrative review and, to some extent, speculative.

We are applying all this changes in the revised version of the manuscript and we will be able to resubmit it tomorrow.

King regards,

Authors

Round 2

Reviewer 1 Report

The work is now ready for publication thanks to the author's modifications.